

# Variations in bacterial diversity and community structure in the sediments of an alkaline lake in Inner Mongolia plateau, China

Jumei Liu[1],[*], Jingli Yu[2],[*], Wantong Si[1], Ge Ding[1], Shaohua Zhang[2], Donghui Gong[3] and Jie Bi[3]

[1] College of Chemistry and Environmental Engineering, Chongqing Key Laboratory of Environmental Materials & Remediation Technologies, Chongqing University of Arts and Sciences, Chongqing, China
[2] Ministry of Education Key Laboratory of Ecology and Resource Use of the Mongolian Plateau, School of Ecology and Environment, Inner Mongolia University, Hohhot, Inner Mongolia Autonomous Region, China
[3] School of Life Science and Technology, Inner Mongolia University of Science and Technology, Baotou, Inner Mongolia Autonomous Region, China
[*] These authors contributed equally to this work.

Corresponding author
Wantong Si, siwt02@163.com

## ABSTRACT

Alkaline lakes are a special aquatic ecosystem that act as important water and alkali resource in the arid-semiarid regions. The primary aim of the study is to explore how environmental factors affect community diversity and structure, and to find whether there are key microbes that can indicate changes in environmental factors in alkaline lakes. Therefore, four sediment samples (S1, S2, S3, and S4) were collected from Hamatai Lake which is an important alkali resource in Ordos' desert plateau of Inner Mongolia. Samples were collected along the salinity and alkalinity gradients and bacterial community compositions were investigated by Illumina Miseq sequencing. The results revealed that the diversity and richness of bacterial community decreased with increasing alkalinity (pH) and salinity, and bacterial community structure was obviously different for the relatively light alkaline and hyposaline samples (LAHO; pH < 8.5; salinity < 20‰) and high alkaline and hypersaline samples (HAHR; pH > 8.5; salinity > 20‰). Firmicutes, Proteobacteria and Bacteriodetes were observed to be the dominant phyla. Furthermore, Acidobacteria, Actinobacteria, and low salt-tolerant alkaliphilic nitrifying taxa were mainly distributed in S1 with LAHO characteristic. Firmicutes, Clostridia, Gammaproteobacteria, salt-tolerant alkaliphilic denitrifying taxa, haloalkaliphilic sulfur cycling taxa were mainly distributed in S2, S3 and S4, and were well adapted to haloalkaline conditions. Correlation analysis revealed that the community diversity (operational taxonomic unit numbers and/or Shannon index) and richness (Chao1) were significantly positively correlated with ammonium nitrogen (r = 0.654, $p < 0.05$; r = 0.680, $p < 0.05$) and negatively correlated with pH (r = −0.924, $p < 0.01$; r = −0.800, $p < 0.01$; r = −0.933, $p < 0.01$) and salinity (r = −0.615, $p < 0.05$; r = −0.647, $p < 0.05$). A redundancy analysis and variation partitioning analysis revealed that pH (explanation degrees of 53.5%, pseudo-F = 11.5, $p < 0.01$), TOC/TN (24.8%, pseudo-F = 10.3, $p < 0.05$) and salinity (9.2%, pseudo-F = 9.5, $p < 0.05$) were the most significant factors that caused the variations in bacterial community structure. The results suggested that alkalinity,

nutrient salt and salinity jointly affect bacterial diversity and community structure, in which one taxon (Acidobacteria), six taxa (Cyanobacteria, Nitrosomonadaceae, *Nitrospira*, *Bacillus*, *Lactococcus* and *Halomonas*) and five taxa (*Desulfonatronobacter*, *Dethiobacter*, *Desulfurivibrio*, *Thioalkalivibrio* and *Halorhodospira*) are related to carbon, nitrogen and sulfur cycles, respectively. Classes Clostridia and Gammaproteobacteria might indicate changes of saline-alkali conditions in the sediments of alkaline lakes in desert plateau.

## INTRODUCTION

Alkaline lakes are a type of saline lake, which contains high concentrations of alkaline salts such as sodium carbonate/bicarbonate, leading to high pH and salinity (*Boros & Kolpakova, 2018*). Despite having a higher osmotic pressure and lower nutrition than freshwater lakes, these saline and alkaline environments surprisingly harbor diverse community of microbes (*Sorokin et al., 2014*; *Zhao et al., 2020*; *Zhang et al., 2023*). Generally, microorganisms can be used as indicators to assess environmental changes owing to their high sensitivity to the environment (*Ma et al., 2022*). Contrarily, variations in bacterial communities in aquatic ecosystems can be driven by a number of environmental factors, including geographic distance, pH, salinity, moisture, soil texture, nutrients (total organic carbon, phosphorus, C/N ratio, nitrate, ammonium, nitrogen *etc.*,) (*Hollister et al., 2010*; *Xiong et al., 2012*; *Liu et al., 2015*; *Baatar et al., 2016*; *Yao et al., 2022*; *Zhang et al., 2023*). Bacterial community structure in rivers/lakes were primarily studied by examining a single parameter: either pH (*Liu et al., 2015*) or salinity (*Zhang et al., 2023*). However, several studies revealed that salinity gradient did not have any direct impact on the bacterial communities in alkaline lake sediments across Tibetan Plateau (*Xiong et al., 2012*), hypersaline lake sediments in southern Texas (*Hollister et al., 2010*) and saline-alkaline soils from different regions of Inner Mongolia (*Zhang et al., 2021b*). Recently, *Yue et al. (2019)* confirmed the interactive effects of salinity and nutrients on microbial diversity and composition in lakes. There are studies regarding influence of environmental factors on microbes in some alkaline/saline lakes. However, it is still unknown which factors lead to variations in indigenous microbial communities in different alkaline lakes and whether there are key microbes that can indicate changes in environmental factors. Any variation in the microbial community can further lead to changes in the metabolic processes affecting the stability of the entire ecosystem (*Calderón et al., 2017*). Therefore, it is necessary to identify the driving factors of bacterial community structure in different ecosystems.

Alkaline lake ecosystems in desert plateaus are characterized by a semiarid or arid climate with high evaporation, scarce precipitation, shallow groundwater level and surface salt accumulation (*Sorokin et al., 2014*). In Ordos plateau of Inner Mongolia alone, there are 65 alkaline lakes with an area of more than 1 km$^2$, and the total area of alkaline lakes is

189 km² (*Qiao, Li & Zeng, 2001*). These alkaline lakes may provide an excellent research field to explore the response of microorganisms to environmental conditions.

Thus, the main goals of this study were (1) to analyze the diversity and composition of the bacterial communities based on Illumina Miseq sequencing and bioinformatics analysis; (2) to identify the main environmental factors driving the variation in the sediment bacterial communities in the alkaline lake; (3) to find whether there are key microbes that can indicate changes in environmental factors in alkaline lakes in desert plateau. This study provides a theoretical basis for future sustainable development of the naturally fragile lake ecological environments in the desert plateau.

## MATERIALS AND METHODS

### Sediment samples and physicochemical analysis

Hamatai Lake is one of the most important lakes in the area with a high concentration of alkalis existing in the forms of carbonate/bicarbonate (*Lu, Xiang & Wen, 2011*). Hamatai Lake is located on the Ordos' desert Plateau in Etuokeqi County, Inner Mongolia (latitude: 39°05′27.78″N to 39°06′35.22″N, longitude: 108°01′37.5″E to 108°02′49.38″E). It is 0.5–0.8 m in depth and spread across the area of about 2.5 km². Hamatai Lake exhibit an east-to-west gradient of increasing alkalinity (pH) and salinity, as well as allochthonous deposition of nutrients. The area is characterized by a semi-arid or arid climate with strong evaporation (2,500–2,700 mm y⁻¹) and scarce precipitation (280–360 mm y⁻¹). Field experiments were approved by School of Life Science and Technology, Inner Mongolia University of Science and Technology, China. According to water color and landscape characteristics of lakeshore, four sediment sites were selected in Hamatai Lake (Fig. S1). Site S1 was located in the eastern side of lake with vegetation coverage but no obvious salt crust at the east lakeshore. Site 2 was in the middle of lake with blue-green water. Two sediments sites S3 (with light-yellow water) and S4 (with deep-yellow water) were selected from the western region of lake with obvious salt crust but no vegetation coverage at the west lakeshore. Three samples were randomly collected from each site. Each parallel sample was mixed thoroughly by five sediment cores. Using sediment sampler (KC mod A ochB; Swedaq, Oslo, Norway), all samples of 0–10 cm sediments were collected in sterile plastic bags and immediately transported to the laboratory in ice-cooled boxes. All samples were divided into two parts. One part was stored at −80 °C for DNA extraction and the other part was air-dried and sieved for sediment physiochemical parameters analysis. Physicochemical characteristics were analyzed as described in previous studies (*Zhang et al., 2023*). The pH was determined using a pH meter (HQ40D; HACH, Ames, IA, USA) with a 1:5 sediment to water solution suspension. Water content (WC) was determined by constant weight method at 105 °C. Salinity was measured using residue drying method. Total organic carbon (TOC) and total nitrogen (TN) were measured using elemental analyzer (Vario EL Cube; Elementar, Langenselbold, Germany). Nitrate nitrogen ($NO_3^-$-N) and ammonium nitrogen ($NH_4^+$-N) were determined using a Smartchem discrete auto analyzer (Smartchem140; Westco, Cerizay, France) after being extracted from sediments using 2 mol L⁻¹ KCl solution and filtered through 0.45 μm filters.

## DNA extraction, PCR amplification, and pyrosequencing

Sediment DNA was extracted by using a fast DNA extraction kit (MP Biomedicals, Solon, OH, USA). The primers 515F (5′-GTGCCAGCMGCCGCGG-3′) and 907R (5′-CCGTCAATTCMTTTRAGTTT-3′) were used to amplify the V4–V5 region of 16S rRNA (*Xiong et al., 2012*). Sequencing of PCR products was carried in Majorbio Bio-Pharm Technology Co. Ltd. (Shanghai, China) using illumina MiSeq platform (illumina, San Diego, CA, USA) and according to the standard protocols.

## Sequence processing and data statistical analysis

Raw fastq files were quality-filtered by Trimmomatic (http://www.usadellab.org/cms/index.php?page=trimmomatic) and then merged by FLASH (http://ccb.jhu.edu/software/FLASH/index.shtml) with the following criteria: (i) The reads were truncated at any site receiving an average quality score <20 over a 50 bp sliding window. (ii) Primers were exactly matched allowing two nucleotide mismatching, and reads containing ambiguous bases were removed. (iii) Sequences whose overlap longer than 10 bp were merged according to their overlap sequence (*Magoč & Salzberg, 2011*; *Bolger, Lohse & Usadel, 2014*). Operational taxonomic units (OTUs) were clustered with 97% similarity cutoff using UPARSE (*Edgar, 2013*). Chimeric sequences were identified and removed using UCHIME (*Edgar et al., 2011*). The diversity indices (*i.e.*, coverage, Chao1, ACE, Shannon, and Simpson index) were estimated by using Mothur (*Schloss et al., 2009*). The taxonomy of each 16S rRNA gene sequence under confidence threshold of 70% was analyzed by RDP Classifier algorithm (http://rdp.cme.msu.edu/) against the Silva (SSU123) 16S rRNA database (*Wang et al., 2007*).

The differences in the sediment physicochemical data, bacterial diversity indices and bacterial communities' relative abundance were detected through one-way analysis of variance using Duncan test under the criteria of the normality and homogeneity of variance ($p > 0.05$). Pearson correlation analysis were performed to analyze correlation factor between environmental factors and microbial species, as well as correlation of the environmental factors themselves (significance level $\alpha = 0.05$). SPSS 25.0 software was used for statistical analysis. Unweight UniFrac distance-based hierarchical clustering tree and principal co-ordinates analysis (PCoA) were built using "vegan" package of R (version 3.0.0). Several functional groups in taxa of family and genus with relatively high abundances were compared using heatmap analysis. The relationships between environmental factors and bacterial community composition were determined by redundancy analysis (RDA) using the software CANOCO (version 5.0) (*Zhang et al., 2023*). The individual contribution of each environmental factor to the variation of bacterial community was also determined by variation partition analysis (VPA) using CANOCO (*Yue et al., 2019*). Monte Carlo test with 999 permutations was used to determine significant difference with variance adjustment. The significance level was $p < 0.05$.

**Table 1 Physicochemical properties of sediment samples ($n = 3$).**

|  | S1 | S2 | S3 | S4 |
|---|---|---|---|---|
| pH | 7.63 ± 0.12c | 9.33 ± 0.12a | 8.90 ± 0.08b | 9.43 ± 0.05a |
| Salinity (‰) (w/w) | 16.05 ± 0.30d | 49.15 ± 0.10a | 22.81 ± 0.79c | 40.62 ± 0.73b |
| WC (%) | 25.25 ± 0.48b | 40.68 ± 0.87a | 14.98 ± 0.27d | 20.19 ± 0.29c |
| TOC (g/kg) | 28.53 ± 0.77b | 30.92 ± 0.26a | 24.43 ± 0.31c | 20.83 ± 0.63d |
| TN (g/kg) | 0.87 ± 0.07b | 1.34 ± 0.03a | 0.72 ± 0.07c | 0.87 ± 0.02b |
| TOC/TN | 33.07 ± 3.52a | 23.02 ± 0.26b | 34.34 ± 2.83a | 23.88 ± 1.34b |
| $NO_3^-$-N (mg/kg) | 0.40 ± 0.06d | 3.17 ± 0.05a | 1.07 ± 0.06c | 1.80 ± 0.02b |
| $NH_4^+$-N (mg/kg) | 3.44 ± 0.40a | 1.97 ± 0.40b | 2.95 ± 0.40a | 1.80 ± 0.23b |

**Note:**
Values were shown as Mean ± SD of triplicate ($n = 3$) data sets. Different letters in a row indicate differences among four sediment sites ($p < 0.05$). WC, water content; TOC, total organic carbon content; TN, total nitrogen content; TOC/TN, ratio of total organic carbon content to total nitrogen content; $NO_3^-$-N, nitrate nitrogen content; $NH_4^+$-N, ammonium nitrogen content.

## Nucleotide sequence accession numbers

The raw sequences of all samples (fastq files) were deposited in Sequence Read Archive of NCBI under accession number SRP132324.

## RESULTS

### Sediment physicochemical properties

Physicochemical properties of the four sites are summarized in Table 1. The sediments were mainly alkaline in nature (pH > 7.0). Salinity ranged from 16.1 to 49.1‰. Sediments of site S1 were relatively low alkaline (pH < 8.5) and hyposaline (salinity < 20‰) (LAHO), while the other three sites consisted of high alkaline (pH > 8.5) and hypersaline (salinity > 20‰) sediments (HAHR). The WC, TOC, TN and $NO_3^-$-N of S2 in the middle of lake were greater than S1, S3 and S4. However, higher TOC/TN and $NH_4^+$-N were observed in S1 and S3.

### Distribution of bacterial community

A total of 95,543 high-quality sequences were gained from the four sites (Table 2). At the 97% OTU level, the coverage per sample was higher than 98.9%, and all rarefaction curves reached to the saturation level (Fig. S2). Shannon and Simpson indices estimated species diversity of sediment microbial communities, while Chao1 and ACE indices estimated species richness of sediment microbial communities (Table 2). S1 samples revealed the most diverse bacterial community, while S4 samples had the simplest bacterial community. Compared to S1 site, the diversity and richness of the bacterial communities in the HAHR sites were observed to be significantly lower ($p < 0.05$, Table 2). Moreover, the diversity and richness of bacterial community within the HAHR sites varied significantly ($p < 0.05$; Table 2). Overall, along the alkalinity (pH) and salinity gradients from the highest to the lowest, the diversity indices showed the opposite trend (Table 2).

**Table 2 Diversity and richness index of bacterial community for sediment samples ($n = 3$).**

| Sources | S1 | S2 | S3 | S4 |
|---|---|---|---|---|
| Reads | 21,172 | 25,662 | 24,211 | 24,498 |
| OTUs | 1,407 ± 21a | 694 ± 29b | 550 ± 67c | 426 ± 59d |
| Distance | 0.03 | 0.03 | 0.03 | 0.03 |
| Coverage | 0.989 ± 0.002 | 0.993 ± 0.003 | 0.994 ± 0.000 | 0.995 ± 0.001 |
| Shannon | 6.045 ± 0.014a | 4.784 ± 0.047b | 3.783 ± 0.115c | 3.178 ± 0.527d |
| Simpson (1/D) | 152.939 ± 13.638a | 56.019 ± 3.245b | 9.759 ± 1.296c | 9.403 ± 4.191c |
| Chao1 | 1,543.602 ± 33.381a | 837.465 ± 34.220b | 746.191 ± 62.451b | 536.671 ± 59.637c |
| ACE | 1,541.759 ± 23.720a | 821.607 ± 12.084b | 708.378 ± 68.267c | 534.875 ± 54.746d |

**Note:**
Different letters in a row indicate differences among four sediment sites ($p < 0.05$).

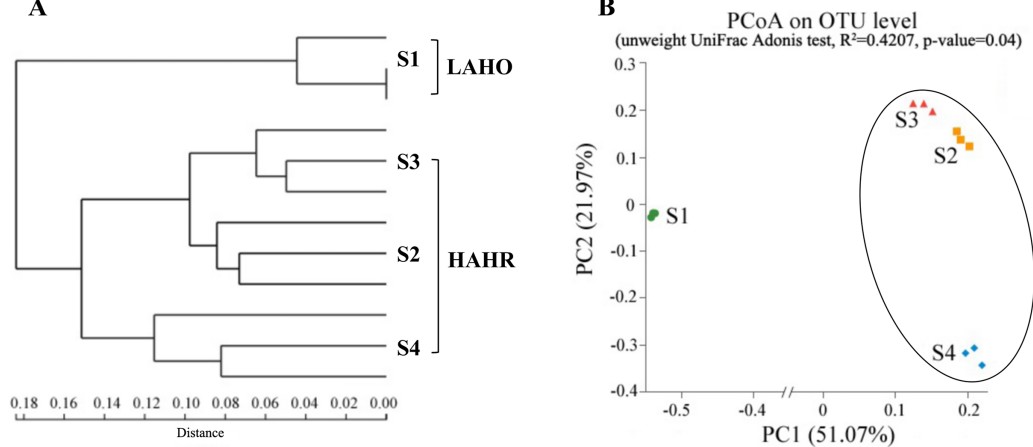

**Figure 1 (A) Hierarchical clustering at the phylum level and (B) PCoA ordination diagrams at OTU level based on the unweight UniFrac distance.** LAHO, relatively light alkaline and hyposaline sediment; and HAHR, high alkaline and hypersaline sediment.

## Composition of bacterial communities

Hierarchical clustering showed that the bacterial communities were clustered into two groups: LAHO (pH: 7.63; salinity: 16.1‰) and HAHR (pH: 8.90–9.43; salinity: 22.8–49.1‰) as shown in Fig. 1A. PCoA analysis also showed that the bacterial community in the site S1 was significantly different from that in the sites S2, S3 and S4 ($R^2 = 0.4207$, $p < 0.05$) (Fig. 1B). Further analyses, the composition of bacterial community in all sites at phylum level were dominated by Firmicutes (relative abundance: S1, 5%; S2, 46%; S3, 16%; S4, 76%; $p < 0.01$), Proteobacteria (relative abundance: S1, 30%; S2, 21%; S3, 28%; S4, 14%; $p < 0.05$), and Bacteriodetes (relative abundance: S1, 7%; S2, 16%; S3, 16%; S4, 3%; $p < 0.01$). Additionally, Acidobacteria (relative abundance: 14%) and Actinobacteria (relative abundance: 10.51%) were mainly present in LAHO rather than HAHR sites, with significant differences between LAHO and HAHR sites ($p < 0.01$). Cyanobacteria (relative abundance: 32%) were mainly found in S3 site, while Nitrospirae were present in S1 site (relative abundance: 1.4%). Additionally, Deinococcus-Thermus

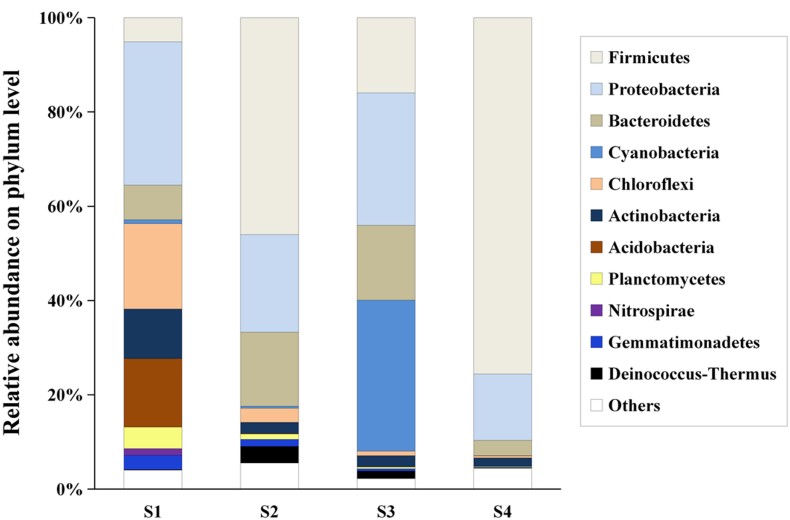

**Figure 2 The composition and relative abundances of the dominant bacterial phyla.** "Others" represents the group of phyla with relative abundance less than 0.1%.

and Gemmatimonadetes were observed in four sites in low abundance (0.1% to 3.55%). Other phyla were also present in lower abundances (2.22% to 5.53%) in all sites (Fig. 2). At class level, a total of 112 taxa were identified across four sites. The dominant classes across all sediments were Bacilli (relative abundance: S1, 0.03%; S2, 10.43%; S3, 0.03%; S4, 62.38%; $p < 0.01$), Clostridia (relative abundance: S1, 1.82%; S2, 34.84%; S3, 12.30%; S4, 12.76%; $p < 0.05$), Gammaproteobacteria (relative abundance: S1, 4.33%; S2, 12.35%; S3, 7.31%; S4, 10.56%; $p < 0.05$), Alphaproteobacteria (relative abundance: S1, 10.55%; S2, 3.79%; S3, 17.24%; S4, 2.29%; $p < 0.05$), Deltaproteobacteria (relative abundance: S1, 5.45%; S2, 4.42%; S3, 3.34%; S4, 1.30%), accounting for more than 72% of the bacterial sequences. Overall, classes Clostridia and Gammaproteobacteria mainly present in HAHR rather than LAHO sites.

Microbial community analysis was performed at a finer level, where the top 50 bacterial taxa from four sites were selected on the basis of relative abundance to compare the microbial community structure in these sites (Fig. S3). The top 50 bacterial taxa at genus level were clustered into two big clusters (cluster A and cluster B). These two big clusters were further divided into eight small clusters. Cluster A1 and cluster A2 were mainly found in the LAHO sediment, while cluster B1 to cluster B6 were present in HAHR sediments. Within these small clusters, salt-tolerant alkaliphilic denitrifying taxa (*Bacillus*, *Lactococcus*, *Halomonas*) and haloalkaliphilic sulfur cycling taxa (Halobacteroidaceae, *Halorhadospira*, *Halanaerobium*, *Alkaliphilus*, *Thioalkalivibrio*, *Dethiobacter*, *Desulfurivibrio*, *Desulfonatronobacter*) were mainly present in the sediments of HAHR sites (Fig. 3). The distribution of these taxa was consistent with the distribution of Firmicutes and Cyanobacteria phyla (Fig. 2). Meanwhile, low salt-tolerant alkaliphilic nitrifying taxa (Nitrosomonadaceae, *Nitrospira*) were mostly present in the LAHO site (Fig. 3), which was consistent with the distribution characteristics of Acidobacteria and

**Figure 3 Relative abundance of main functional groups related to nitrogen and sulfur cycles in the top 50 bacterial taxa at genus level.** The height of the bar chart in each cell shows the relative abundance of a family/genus in a sample. LAHO, relatively light alkaline and hyposaline sediment; and HAHR, high alkaline and hypersaline sediment.

Actinobacteria phyla (Fig. 2). In addition, the distribution of different bacterial taxa was also influenced by environmental factors as microbes show preference to specific environmental factors (Table 3). Except for phyla Bacteroidetes, Cyanobacteria and Deinococcus-Thermus, other seven phyla as well as classes Clostridia (r = 0.695, $p < 0.05$; r = 0.860, $p < 0.01$), Gammaproteobacteria (r = 0.722, $p < 0.01$; r = 0.809, $p < 0.01$), family Nitrosomonadaceae (r = −0.930, $p < 0.01$; r = −0.676, $p < 0.05$), genera *Nitrospira* (r = −0.953, $p < 0.01$; r = −0.694, $p < 0.05$), *Halanaerobium* (r = 0.749, $p < 0.01$; r = 0.870, $p < 0.01$) were both significantly related to pH and salinity (Table 3). Moreover, phylum Gemmatimonadetes (r = −0.835, $p < 0.01$) and genus *Desulfonatronobacter* (r = 0.653, $p < 0.05$) were only correlated with pH (Table 3). Thus, differences in bacterial community composition were observed at phylum, class and genus levels, which showed significant correlation with saline-alkali condition ($p < 0.05$). These results indicate that the heterogeneous sediment environment of alkaline lakes harbors unique microorganisms.

## Environmental factors affecting the diversity, richness, and bacterial community composition

Pearson analysis (Table 3) showed that OTU numbers and Chao1 were negatively correlated with pH (r = −0.924, $p < 0.01$; r = −0.933, $p < 0.01$) and salinity (r = −0.615, $p < 0.05$; r = −0.647, $p < 0.05$), and positively correlated with $NH_4^+$-N (r = 0.654, $p < 0.05$; r = 0.680, $p < 0.05$), respectively. Moreover, Shannon index was negatively (r = −0.800, $p < 0.01$) and positively (r = 0.755, $p < 0.05$) correlated with pH and TOC, respectively (Table 3). Also, Pearson correlation analysis showed a significant positive correlation between pH and salinity (r = 0.838, $p < 0.01$). Both pH and salinity were negatively correlated with TOC/TN (r = −0.650, $p < 0.05$; r = −0.870, $p < 0.01$) and $NH_4^+$-N

**Table 3 Pearson correlation between community diversity, richness, the dominant phyla/class and functional groups and environmental variables.**

| Diversity/Function groups (at phylum/family/Genus level) | | pH | Salinity | WC | TOC | TN | TOC/TN | $NO_3^-$-N | $NH_4^+$-N |
|---|---|---|---|---|---|---|---|---|---|
| Community diversity and richness | OTU numbers | **−0.924**\*\* | **−0.615**\* | 0.186 | 0.546 | −0.003 | 0.403 | −0.538 | **0.654**\* |
| | Shannon | **−0.800**\*\* | −0.422 | 0.435 | **0.755**\*\* | 0.254 | 0.274 | −0.284 | 0.572 |
| | Chao1 | **−0.933**\*\* | **−0.647**\* | 0.169 | 0.570 | −0.021 | 0.453 | −0.552 | **0.680**\* |
| Main phyla/class | Firmicutes | **0.794**\*\* | **0.803**\*\* | 0.147 | −0.449 | 0.292 | **−0.767**\*\* | **0.619**\* | **−0.847**\*\* |
| | Proteobacteria | **−0.734**\*\* | **−0.759**\*\* | −0.151 | 0.396 | −0.267 | **0.679**\* | **−0.588**\* | **0.778**\*\* |
| | Bacteroidetes | 0.135 | 0.092 | 0.278 | 0.515 | 0.292 | 0.164 | 0.328 | 0.151 |
| | Cyanobacteria | 0.046 | −0.419 | **−0.607**\* | −0.246 | **−0.576**\* | **0.626**\* | −0.305 | 0.316 |
| | Chloroflexi | **−0.929**\*\* | **−0.635**\* | 0.117 | 0.460 | −0.075 | 0.412 | **−0.592**\* | **0.659**\* |
| | Actinobacteria | **−0.926**\*\* | **−0.674**\* | 0.019 | 0.368 | −0.134 | 0.394 | **−0.646**\* | **0.705**\* |
| | Acidobacteria | **−0.948**\*\* | **−0.695**\* | 0.006 | 0.357 | −0.186 | 0.462 | **−0.674**\* | **0.683**\* |
| | Planctomycetes | **−0.916**\*\* | **−0.604**\* | 0.180 | 0.525 | −0.026 | 0.414 | −0.541 | **0.652**\* |
| | Nitrospirae | **−0.952**\*\* | **−0.693**\* | −0.002 | 0.348 | −0.189 | 0.455 | **−0.676**\* | **0.641**\* |
| | Deinococcus-thermus | 0.418 | 0.538 | **0.626**\* | **0.583**\* | **0.706**\* | −0.366 | **0.698**\* | −0.242 |
| | Gemmatimonadetes | **−0.835**\*\* | −0.441 | 0.394 | **0.713**\*\* | 0.217 | 0.265 | −0.338 | 0.528 |
| | Gammaproteobacteria | **0.722**\*\* | **0.809**\*\* | 0.420 | −0.044 | 0.555 | **−0.726**\*\* | **0.787**\*\* | **−0.723**\*\* |
| | Clostridia | **0.695**\* | **0.860**\*\* | **0.743**\*\* | 0.420 | **0.817**\*\* | **−0.616**\* | **0.965**\*\* | **−0.628**\* |
| Low salt-tolerant alkaliphilic nitrifying taxa | Nitrosomonadaceae | **−0.930**\*\* | **−0.676**\* | −0.005 | 0.328 | −0.164 | 0.398 | **−0.665**\* | **0.588**\* |
| | *Nitrospira* | **−0.953**\*\* | **−0.694**\* | −0.003 | 0.348 | −0.190 | 0.456 | **−0.676**\* | **0.641**\* |
| Salt-tolerant alkaliphilic denitrifying taxa | *Bacillus* | 0.526 | 0.46 | −0.193 | **−0.679**\* | −0.04 | −0.513 | 0.210 | **−0.640**\* |
| | *Lactococcus* | 0.541 | 0.469 | −0.192 | **−0.688**\* | −0.037 | −0.523 | 0.220 | **−0.640**\* |
| | *Halomonas* | 0.410 | −0.054 | **−0.617**\* | **−0.699**\* | −0.520 | 0.147 | −0.086 | −0.106 |
| Haloalkaliphilic sulfur cycling taxa | *Halanaerobium* | **0.749**\*\* | **0.870**\*\* | 0.399 | −0.226 | 0.530 | **−0.879**\*\* | **0.746**\*\* | **−0.765**\*\* |
| | *Halorhodospira* | 0.399 | 0.314 | −0.189 | **−0.638**\* | −0.083 | −0.467 | 0.136 | −0.401 |
| | Halobacteroidaceae | 0.431 | **0.733**\*\* | **0.885**\*\* | **0.667**\* | **0.908**\*\* | −0.555 | **0.867**\*\* | −0.453 |
| | *Alkaliphilus* | 0.506 | 0.422 | −0.210 | **−0.717**\*\* | −0.095 | −0.534 | 0.174 | **-0.584**\* |
| | *Thioalkalivibrio* | 0.461 | **0.649**\* | **0.771**\*\* | **0.647**\* | **0.818**\*\* | −0.438 | **0.820**\*\* | −0.398 |
| | *Dethiobacter* | 0.493 | **0.748**\*\* | **0.840**\*\* | **0.625**\* | **0.885**\*\* | −0.540 | **0.893**\*\* | −0.454 |
| | *Desulfurivibrio* | 0.406 | **0.737**\*\* | **0.924**\*\* | **0.702**\* | **0.942**\*\* | −0.566 | **0.874**\*\* | −0.428 |
| | *Desulfonatronobacter* | **0.653**\* | 0.455 | 0.110 | 0.031 | 0.260 | −0.217 | 0.571 | −0.354 |

Note:
Asterisk indicated significant correlation (\*$p < 0.05$; \*\*$p < 0.01$). Significant differences at $p < 0.05$ are shown in bold font.

(r = −0.802, $p < 0.01$; r = −0.835, $p < 0.01$), respectively. Overall, the effects of pH on microbial diversity and richness were consistent with the effects of salinity, but opposite to the effects of $NH_4^+$-N. These results suggested that pH, salinity, TOC and $NH_4^+$-N were the key environmental factors influencing the diversity and richness of bacterial community.

RDA analysis was performed to test the relationship between environmental factors and bacterial community structure based on phylum level. The Monte Carlo test for the first (pseudo-F = 4.0, $p < 0.01$) and all canonical axes (pseudo-F = 29.5, $p < 0.01$) were found to be significant which indicates that these sediment parameters are important in explaining the bacterial community composition. RDA1 and RDA2 explained 57.03% and 35.74% of

**A**

**B**

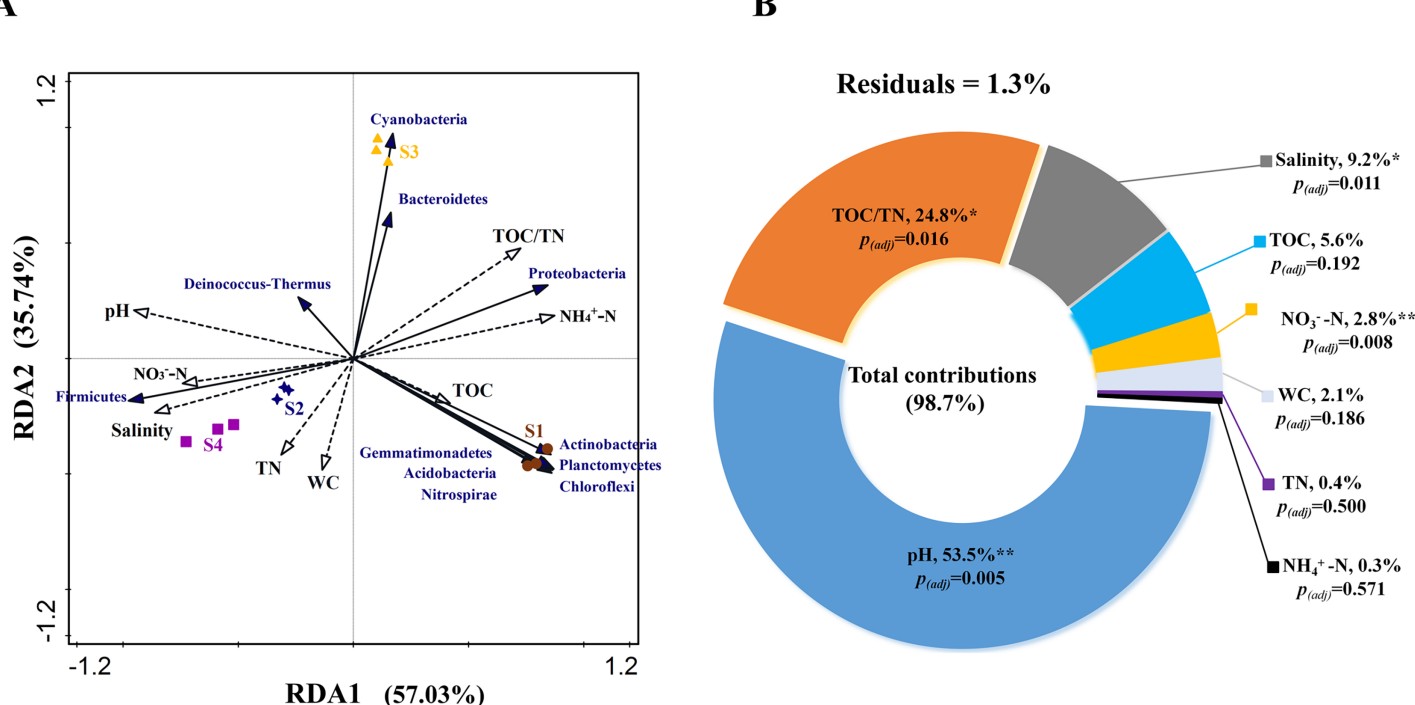

**Figure 4 (A) Relationship between bacterial community and environmental factors by redundancy analysis (RDA) and (B) contributions of environmental factors to form bacterial community structure by variation partitioning analysis (VPA).** TOC, total organic carbon content; TN, total nitrogen content; WC, water content; $NO_3^-$-N, nitrate nitrogen content; and $NH_4^+$-N, ammonium nitrogen content. Significant correlation, $**p < 0.01$ and $*p < 0.05$; non-significant correlation, $p > 0.05$; and $p_{(adj)}$ is value for false discovery rate to $p$ value correction control.

the variation, respectively (Fig. 4A). The VPA results indicated that the total contribution of eight environmental factors to the variation in bacterial community was 98.7% (Fig. 4B). Among all the evaluated environmental factors, pH was the most important factor to influence the community (contribution: 53.5%; pseudo-F = 11.5, $p_{(adj)} < 0.01$), followed by TOC/TN and salinity (explanation degrees: 24.8% and 9.2%, respectively; pseudo-F = 10.3, $p_{(adj)} < 0.05$; pseudo-F = 9.5, $p_{(adj)} < 0.05$) (Fig. 4B). $NO_3^-$-N showed a small contribution to the total variations in bacterial community structure (explanation degree: 2.8%; pseudo-F = 7.3, $p_{(adj)} < 0.01$) (Fig. 4B).

## DISCUSSION

An alkaline lake with extremely high pH and salinity has always been a unique aquatic ecosystem in desert plateau, which plays an important role in the regional hydrological cycle (*Boros & Kolpakova, 2018*). Microbes, the most abundant and diverse life form in the alkaline lake, play a critical role in geochemical cycle of elements. Meanwhile, they are also closely related to environment factors of their habitat, such as salts, alkalis and nutrients. In this study, Hamatai Lake showed high heterogeneity of salts, alkalis and nutrients (Table 1), and bacterial community structure was clearly different across the lake (Fig. 2).

## Effects of environmental factors on bacterial diversity and community structure

Microbial diversity is a good biological indicator of the ecological risks and changes in ecological function for saline-alkaline stress (*Zhao et al., 2020*; *Athen, Dubey & Kyndt, 2021*). Some studies reported that pH is the primary factor that drives sediment/soil microbial diversity and community structure in various water body/soil types (*Xiong et al., 2012*; *Lanzén et al., 2015*; *Liu et al., 2015*; *Huang et al., 2016*) even at the continental scale (*Lauber et al., 2009*). Salinity has also been reported as the major factor regulating microbial communities in saline water of marine (*Pavloudi et al., 2016*) or lakes (*Baatar et al., 2016*; *Liu et al., 2018*; *Lindsay et al., 2019*; *Zhang et al., 2023*), in soils of desert ecosystem (*Zhang et al., 2019*) and even many other environments at global scale (*Lozupone & Knight, 2007*). In this study, the most crucial environmental factors affecting bacterial richness (Chao1 index) and diversity (Shannon index, OTU numbers) was pH, followed by salinity, $NH_4^+$-N and TOC content (Table 3). pH was negatively correlated with Chao1 index ($r = -0.933$, $p < 0.01$), Shannon index ($r = -0.800$, $p < 0.01$) and OTU numbers ($r = -0.924$, $p < 0.01$), while salinity was negatively correlated with Chao1 index ($r = -0.647$, $p < 0.05$) and OTU number ($r = -0.615$, $p < 0.05$) (Table 3). These observations were in agreement with the previous findings in saline/alkaline lakes (*Xiong et al., 2012*; *Yang et al., 2016*; *Lindsay et al., 2019*). Moreover, positive correlation of bacterial community richness and diversity was observed with $NH_4^+$-N ($r = 0.654$, $p < 0.05$; $r = 0.680$, $p < 0.05$) and TOC content ($r = 0.755$, $p < 0.05$), respectively (Table 3). This may be attributed to the fact that microbial growth and biomass can be facilitated *via* organic matter addition in high pH and salinity soils (*Kamble et al., 2014*). Organic matter decomposition provides rich N and P, and labile substrates for mixotrophs leading to increased bacterial diversity (*Wu et al., 2017*).

According to RDA and VPA analysis also, pH was the most important factor affecting the bacterial community structure, followed by TOC/TN and salinity (Figs. 4A and 4B). This may be because pH can lead to stronger environmental filtration and the subsequent species replacement occurring under alkalinity and salinity stress (*Huang et al., 2016*; *Wang et al., 2022*). Generally, microbes have their optimal pH and salinity for growth (*Fernández-Calviño et al., 2011*). Beyond the optimal range of pH, growth of some microbes may reduce even to the point where they are unable to survive. On the contrary, growth of microbes belonging to certain tolerant taxa may be enhanced, leading to obvious variations in community composition (*Lauber et al., 2009*). Moreover, pH integrating other factors, such as nutrient availability, C/N ratio, moisture, and salinity, can drive sediment bacterial community structure (*Liu et al., 2015*; *Yao et al., 2022*). In the present study, TOC/TN ratio and salinity were also found to influence the community composition (Figs. 4A and 4B), which is in agreement with the previous studies carried out for hypersaline or saline sediments (*Hollister et al., 2010*; *Yang et al., 2016*). In addition, $NO_3^-$-N showed a small contribution to the total variations in bacterial community structure (Fig. 4B). The $NO_3^-$-N content may affect the communities of denitrifying and anammox bacteria in sediments (*Zhang et al., 2021a*).

## Unique bacterial community composition and driving factors

Despite harboring a high biodiversity, alkaline lakes are often considered as extreme aquatic environments with high alkalinity and salinity (*Zhao et al., 2020*; *Hou et al., 2022*). Presence of some tolerant microbes, the relationships between microbial communities and various environmental variables were often revealed for such lakes (*Sorokin & Chernyh, 2017*; *Yue et al., 2019*). In this study, change in bacterial community composition was shown in response to the alkalinity and salinity and various sediment physicochemical properties (Table 3). Firmicutes was the dominant phylum in S2 and S4 sites with high alkaline and hypersaline conditions (Fig. 2). Firmicutes phylum and Clostridia class were both positively related to pH ($r = 794$, $p < 0.01$; $r = 0.695$, $p < 0.05$) and salinity ($r = 0.803$, $p < 0.01$; $r = 0.860$, $p < 0.01$) but negatively related to TOC/TN ($r = −0.767$, $p < 0.01$; $r = −0.616$, $p < 0.05$) (Table 3); whereas Proteobacteria was positively correlated with TOC/TN ratio ($r = 0.679$, $p < 0.05$) (Table 3). The TOC/TN ratio was found to be relatively high in the present study (averaging 25.58). These results suggest that a higher TOC/TN ratio would mitigate the alkalinity and salinity constraints for specific microorganisms, which was previously confirmed by *Yue et al. (2019)*. Moreover, Proteobacteria phylum have been reported to be widely distributed in aquatic ecosystems, while dominance of Betaproteobacteria class was observed in freshwater lake (*Liu et al., 2015*; *Yue et al., 2019*). On the other hand, Gammaproteobacteria class were reported to be more abundant in hypersaline lake (*Liu et al., 2018*). In this study, Gammaproteobacteria class had a significant positive relationship with pH ($r = 0.772$, $p < 0.01$) and salinity ($r = 0.809$, $p < 0.01$) but a negative relationship with TOC/TN ($r = −0.726$, $p < 0.01$) (Table 3). These bacteria are more dominating in low-nutrient environment, and they can adapt to saline-alkaline environment. Therefore, they have been frequently detected in soda/saline lake (*Lin et al., 2017*; *Zhao et al., 2020*). Chloroflexi, Actinobacteria, Acidobacteria, Planctomycetes and Nitrospirae phyla were observed to be more sensitive to saline-alkaline stress showing a significant negative correlation with pH ($r = −0.929$, $p < 0.01$; $r = −0.926$, $p < 0.01$; $r = −0.948$, $p < 0.01$; $r = −0.916$, $p < 0.01$; $r = −0.952$, $p < 0.01$) and salinity ($r = −0.635$, $p < 0.05$; $r = −0.674$, $p < 0.05$; $r = −0.695$, $p < 0.05$; $r = −0.604$, $p < 0.05$; $r = −0.693$, $p < 0.05$), respectively (Table 3). The relative abundance of these phyla in high alkaline and hypersaline sediment significantly decreased (Fig. 2), which is consistent with previous studies in saline sediment (*Baatar et al., 2016*; *Yang et al., 2016*). These bacteria may become dormant under high saline-alkali environment in which nutrients are relatively scarce. Like pH, salinity also acts as a powerful environmental filter for building bacterial activity in lake ecosystems; salt-tolerant groups in hypersaline communities are discovered through high environmental selectivity and may require less dormancy to thrive and survive (*Aanderud et al., 2016*). Therefore, sediment alkalinity, salinity and nutrients may regulate bacterial community structure and composition by affecting the abundance of these dominant phyla or classes.

Microbial species active in the carbon, nitrogen and sulfur cycles are widely distributed in soda/alkaline lake. For carbon cycling, methane-oxidizing bacteria are found abundantly in saline-alkaline lakes and the diversity of methanotrophs decreases with
increasing salinity (*Zhang et al., 2023*). *Sorokin et al. (2014)* also reported that carbon and nitrogen cycles are partly inhibited in soda lakes due to the inability of methanotrophs and ammonia oxidizers to grow at high salinity. In this study, methanotrophs and methanogens were not found or their abundance was below the level of detection. However, Acidobacteria were more abundant in S1 site (Fig. 2). *Wilmoth et al. (2021)* reported that Acidobacteria were involved in the metabolism of C1 compound in peat, such as the production of methanogenic substrates $H_2$ and $CO_2$.

For nitrogen cycling, Cyanobacteria are found abundantly in alkaline lakes which act as nitrogen fixing microbes (*Hou et al., 2022*). *Sorokin et al. (2014)* reported that Cyanobacteria are dominant at medium salinity whereas green algae can survive at higher saline conditions. In this study, haloalkaliphilic Cyanobacteria mainly existed in S3 with relatively medium salinity (salinity: 22.8‰) (Fig. 2), and were significantly related to higher TOC/TN (r = 0.626, *p* < 0.05) (Table 3). Previous study reported that Cyanobacteria can balance the extracellular high pH by synthetizing active sodium proton antiporters. This process requires sufficient sodium, and sodium deficiency may lead to the rapid death of Cyanobacteria in extremely alkaline environments (*Hou et al., 2022*). $Na^+$ was the dominant cation in S3 in this study (Fig. S4). Thus, the medium salinity and sufficient sodium carbonate/bicarbonate may have provided a good habitat for the survival and flourish of Cyanobacteria, which also created unique bio-environmental conditions for *Spirulina* (an alkaliphilic Cyanobacteria) cultivation, even the commercial *Spirulina* (*Qiao, Li & Zeng, 2001*; *Lu, Xiang & Wen, 2011*). In addition, Nitrosomonadaceae (ammonia oxidizing bacteria) and *Nitrospira* affiliated with Nitrospiraceae (nitrite oxidizing bacteria) appeared in LAHO sediments (Fig. 3), which may be attributed to the low diversity of ammonia-oxidizing bacteria and the inhibition of nitrification at high pH and salinity environment (*Sorokin et al., 2014*; *Osborne, Bernot & Findlay, 2015*). In this study, they are significantly negatively correlated with pH (r = −0.930, *p* < 0.01; r = −0.953, *p* < 0.01) and salinity (r = −0.676, *p* < 0.05; r = −0.694, *p* < 0.05) (Table 3). Whereas, the salt-tolerant alkaliphilic genera such as *Marinobacter*, *Halomonas*, *Bacillus* and *Lactococcus* can produce nitrogen oxides in relatively higher salinity habitats, due to their denitrification ability (*Shapovalova et al., 2009*; *Yang et al., 2016*). *Halomonas*, *Bacillus* and *Lactococcus* were been detected in this study (Fig. 3), and positively related to pH or salinity but not significantly (Table 3).

The microbial sulfur cycle appears to be very active in the soda lakes. *Zhao et al. (2020)* found that approximately half of the abundant metagenome assembled genomes had the potential to drive dissimilatory sulfur cycling in soda-saline lakes. Sulfur reducing bacteria (SRB) and sulfur oxidizing bacteria (SOB) are commonly present in these environments and lead to sulfur cycling (*Sorokin et al., 2014*). As shown in Fig. 3 and Fig. S3, HAHR sediments were rich in *Desulfonatronobacter acetoxydans*, *Dethiobacter alkaliphilus* and *Desulfurivibrio alkaliphilus*, which have been reported to be the representative bacteria of SRB (*Sorokin et al., 2008*; *Sorokin, Chernyh & Poroshina, 2015*). These bacteria were positively correlated with pH (r = 0.653, *p* < 0.05) or salinity (r = 0.748, *p* < 0.01; r = 0.737, *p* < 0.01) in this study (Table 3). The genus *Thioalkalivibrio*, a representative of SOB was best adapted to haloalkaline conditions (*Sorokin et al., 2014*; *Mu et al., 2016*), also been

detected in HAHR sediments (Fig. 3), was positively correlated with salinity (r = 0.649, $p < 0.05$) in this study (Table 3). Additionally, *Halorhodospira* (purple sulfur bacteria) was important members of sulfur cycle, been found in S4 site (Fig. 3).

Overall, the changes of environmental factors in this study will affect the microbes associated with carbon, nitrogen and sulfur cycles, such as Acidobacteria, Cyanobacteria, Nitrosomonadaceae, *Nitrospira*, *Halomonas*, *Bacillus*, *Lactococcus*, *Desulfonatronobacter*, *Dethiobacter*, *Desulfurivibrio*, *Thioalkalivibrio* and *Halorhodospira*. Furthermore, it will be crucial to focus on the changes of environmental factors leading to the changes of unique carbon, nitrogen and sulfur microbial community structure in alkaline lakes, which may affect microbially-mediated carbon, nitrogen and sulfur cycle of alkaline lakes in the future.

## CONCLUSIONS

This study investigated the sediment bacterial community diversity and driving force of community assembly in an alkaline lake in desert plateau. Firmicutes, Proteobacteria and Bacteriodetes were the dominant bacterial phyla. The diversity and richness of bacterial communities gradually decreased with the rise in pH and salinity. The dominance of salt- and alkali-tolerance bacteria (*e.g.*, Firmicutes, Clostridia, Gammaproteobacteria, salt-tolerant alkaliphilic denitrifying taxa and haloalkaliphilic sulfur cycling taxa) significantly increased with the rise in pH and salinity, whereas low salt-tolerant alkaliphilic nitrifying taxa significantly decreased. The pH and salinity have an obvious effect on the distribution of bacterial community. However, nutrients (TOC or TOC/TN) may mitigate the alkalinity and salinity constraints for some microbes. In the bacterial community driven by environmental factors, one taxon (*Acidobacteria*), six taxa (Cyanobacteria, Nitrosomonadaceae, *Nitrospira*, *Bacillus*, *Lactococcus* and *Halomonas*) and five taxa (*Desulfonatronobacter*, *Dethiobacter*, *Desulfurivibrio*, *Thioalkalivibrio* and *Halorhodospira*) are related to carbon, nitrogen and sulfur cycles, respectively. The classes, Clostridia and Gammaproteobacteria might indicate changes of saline-alkali conditions in the sediments of alkaline lakes. Our study not only helps to understand the relationships between microbes, saline-alkali, and nutrient, but also provides a scientific basis for future sustainable development of the naturally fragile lake ecological environments in the desert plateau.

### Funding

The project was supported by the National Natural Science Foundation of China (No. 42103078; 31660151), and the Natural Science Foundation of Chongqing, China (No. CSTB2022NSCQ-MSX1300, CSTC2020jcyj-msxmX1011). The funders had no role in study design, data collection and analysis, decision to publish, or preparation of the manuscript.

## Grant Disclosures
The following grant information was disclosed by the authors:
National Natural Science Foundation of China: 42103078, 31660151.
Natural Science Foundation of Chongqing, China: CSTB2022NSCQ-MSX1300, CSTC2020jcyj-msxmX1011.

## Competing Interests
The authors declare that they have no competing interests.

## Author Contributions
- Jumei Liu conceived and designed the experiments, analyzed the data, prepared figures and/or tables, authored or reviewed drafts of the article, and approved the final draft.
- Jingli Yu analyzed the data, prepared figures and/or tables, authored or reviewed drafts of the article, and approved the final draft.
- Wantong Si conceived and designed the experiments, analyzed the data, authored or reviewed drafts of the article, and approved the final draft.
- Ge Ding analyzed the data, prepared figures and/or tables, and approved the final draft.
- Shaohua Zhang performed the experiments, prepared figures and/or tables, and approved the final draft.
- Donghui Gong conceived and designed the experiments, prepared figures and/or tables, and approved the final draft.
- Jie Bi performed the experiments, analyzed the data, prepared figures and/or tables, authored or reviewed drafts of the article, and approved the final draft.

## Field Study Permissions
The following information was supplied relating to field study approvals (*i.e.*, approving body and any reference numbers):

Field experiments were approved by School of Life Science and Technology, Inner Mongolia University of Science and Technology, China.

## DNA Deposition
The following information was supplied regarding the deposition of DNA sequences:

The raw sequences of all samples are available in the Sequence Read Archive of NCBI: SRP132324; PRJNA433101.

https://trace.ncbi.nlm.nih.gov/Traces/?study=SRP132324.

## Data Availability
The raw measurements are available in the Supplemental Files.

## Supplemental Information
Supplemental information for this article can be found online at http://dx.doi.org/10.7717/peerj.15909#supplemental-information.

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
