# Peer review of "Variations in bacterial diversity and community structure in the sediments of an alkaline lake in Inner Mongolia plateau, China"

_PeerJ, doi:10.7717/peerj.15909_

## Round 0.1 · original submission · Major Revisions

The reviews of your submission reflect the diverging thoughts on the content. Overall the experimental design and execution meet standards. However, the rationale and discussion of the data need improvement. The major concerns seem to be related to the somewhat descriptive nature of the submission. This is of itself isn't bad as pointed out by reviewers, but there were major issues with the text.
Please revise according to reviewers comments where possible and contextualize your study appropriately and justify the need for just the descriptive nature of the current submission,

Reviewer 1 ·

Basic reporting

This work aims at describing the bacterial community within an alkaline and saline lake in Ordos’ desert plateau of inner Mongolia. In order to do so, they sampled 4 points along the lake, following an environmental gradient of increasing salinity/pH. They have provided raw data, deposited in the SRA website. Figures and tables have good quality and are well described in the text, coinciding with the results.
About the writing and language, I would like to praise the grammar, that is overall good. Even though the English is well written, the text lacks fluidity and a sense of continuity among sentences. Over the entire manuscript, sentences sound like collages of disconnected informations, hampering the reading experience. This major drawback is among one of the main reasons for my final decision.
My second major point of concern is about the extensive use of references and informations that are outside of the scope of the paper (defined by methodology and results). For example, while this work is a description of communities in an environmental gradient (which should focus the introduction/discussion on community assembly and modulation of community structure by environmental variables), the introduction spans from genome and metagenomes of halophytic microbes, mechanisms used by microorganisms to sustain osmotic pressure and climate change (which is also stated as one of the conclusions and important topics of the work, but the authors did not provided how and why their studied samples link with anthropogenic effects). This made the text extensive, without improving the quality of the work (also defocusing from their results and real conclusions). This problem is also seem during the discussion. The conclusion is also overall broad, going further than the results support.
The third major point is about the lack of originality and novelty of the work. While the authors state that “the effects of environmental factors on the diversity and composition
of bacterial community in the sediment of alkaline lake in semiarid region are still unclear”, descriptive works on saline and alkaline environments are very common (as cited in the text), and the role of pH and salinity modulating microbial communities is well described (e.g., https://doi.org/10.1038/nature24621, https://doi.org/10.1128/AEM.00335-09, https://doi.org/10.1038/s41396-018-0313-8). What is lacking, and it is at this point novelty in microbial ecology are the discovery of mechanisms underlying these well described patterns. Even though descriptive works are still valid, the number of samples (3 samples and 4 points in a gradient) is not much to unravel precisely patterns. If relevant in some way that I am not aware, authors should make a bigger effort in order to improve the clarity of the importance of this work (how and why).

Experimental design

This works presents a sound and reliable methodology, in technical terms. I was unable to find anything wrong on how they conducted their experiments. Even though, as stated in the basic reporting by me, the goal of this research is not novel nor highly relevant, and it was not well described and explored in the text. Their objectives, according to the last paragraph of the introduction were: (1) to analyze the diversity and composition of the bacterial communities based on Illumina Miseq sequencing and bioinformatics analysis; 2) identify the main environmental factors driving the variation in the sediment bacterial communities in the alkaline lake of Inner Mongolia Plateau. This is a rather descriptive work, which, in my point of view, fails to provide insights into any novel mechanisms or a deeper understanding of microbial communities. Another disadvantage is the fact that they work with 4 points (3 samples only). This is a low number of samples, and for environmental samples, they provide lots of factors that increase the difficult to reliably state which changes are responsible by what factors.

Validity of the findings

The methodology and analysis seems sound and reliable. Further points were elaborated above.

Additional comments

All the important points were discussed above.

·

Basic reporting

1. Basic Reporting
a. The general overview of the manuscript was easy to follow, and the narrative provided the necessary foundational/technical information to establish knowledge of the subject matter. It was an interesting examination of relatively understudied system that, due to the increased anthropogenic activities, are likely to experience considerable environmental pressure in the future. The reasoning behind this work is well laid out and the data provide a clear foundation for future studies of these types of systems. While there are some issues (as per my comments below), I think this was a worthwhile study and I think it does deserve follow-up (including more data points).
There were some grammatical concerns that need to be addressed (see general comments for specific examples, but plural/singular nouns and article use are most prevalent) and it is suggested that the manuscript is carefully parsed for these types of grammatical errors.
b. There were a number of instances in which concepts were not clearly developed or described in ample detail (see general comments). There were some statements that could not be supported by the references cited, or that did not include sufficient support from the literature (see general comments).
c. All graphics were appropriate and raw data was provided with the supplemental materials.
d. There were no clearly stated hypotheses or predictions. Due to the observational nature of this study I recognize that these may not be as fundamental given the specificity of the subject matter (alkaline lakes); however, there is a substantive body of work on salinity and pH that could have provided direction.

Experimental design

2. Experimental Design
a. This work falls within the aims and scope of the journal.
b. Research objectives are relevant and meaningful. The knowledge gap is addressed, although the wording should be reviewed and reconsidered.
c. Appropriate rigor and technical/ethical standards.
d. Some of the methods used lacked the necessary depth to be repeatable, however this could be remedied with additional information. Methods are appropriate for this study.

Validity of the findings

3. Validity of the findings
a. Check.
b. Data have been provided and are sufficient for the current study.
c. Conclusions were mostly well-stated and reflective of the results/data (see general comments). Causative factors for observations can be inferred from the data, but as this is an observational study it must be adequately contextualized—which it was for the most part (addressed in comments).

Additional comments

4. General Comments
a. L26-28: This sentence can be either separated into two separate sentences or rearranged to improve clarity. Also, see second word.
b. L29: Lake needs to be plural. This issue occurs several times throughout the manuscript. If this work is intended to provide data that can be used to make inferences about alkaline lakes in general, the “alkaline lake” requires an added “s” at the end. If this is solely about a single alkaline lake, then it is best to refer to the lake by name or use the correct article when referencing said lake. Similar issues are found on lines 55, 60, 62, 75,
c. L65-67: I would suggest combining this sentence with the next sentence to reduce redundancies.
d. L77-79: This statement should be reconsidered as there has been ample evidence of microbial resilience when faced with slight disturbance (see Shade et al., 2012; Philippot et al., 2021; Walker et al., 2022) Moreover, the Luo et al (2022) paper may not be the best reference for this statement as that study was focused on rare-earth elements and heavy metals, each of which can have dramatic effects on community composition (see Custodio et al., 2022). The Sorkin et al. (2014) is not really speaking to whether changes in diversity are indicative of changes to ecosystem functionality. It is merely examining the microbial diversity and microbially-mediated functions of soda lakes. There are articles that point to bacteria as indicators of environmental change (for instance, see Ma et al., 2022 and the references therein). Also, was water data (e.g., temperature) collected at the water-sediment interface?
e. L112-114: This sentence is awkward, in part due to article placement (e.g., “the” human activities, “the” natural fragility).
f. L122: What exactly is meant by “electricity and sandy land” here? Could this be worded better or perhaps described in greater detail?
g. L129-133: Should you state that changes are occurring prior to providing the data necessary to make this case? Also, “ultimately lead to changes in potential ecological functions” is not clear. Is this stating that the ecological functions that could potentially occur is being altered? The words “ultimately” and “potential” may be misplaced? “Changes in functional potential” and the addition of some predicted directionality to that change (e.g., reduction or increase in functional potential) might be helpful.
h. L136-137: Choose one and stick with it: Lake Hamatai or Hamatai Lake?
i. L140: “Lake”; either an inclusion of an article or just use the name of the lake.
j. L145: What landscape characteristics were considered and why were they chosen? Does this reflect elevated human activity or interactions with the lake?
k. L147-150: What significance did water coloration have in this context? Was there a specific biochemical characteristic that you were attributing to the observed coloration? Could this be expanded on here?
l. L153: How was the sediment separated? Given the potential for microscale environmental gradients to exist (see Wang et all., 2016), how sediment was partitioned could impact results.
m. L174-179: References for programs? Program-specific settings that were used?
n. L224-226: In figure 1B it states that the R2 and p values are from adonis—essentially a permutational MANOVA. Was this run on community abundance data or from the PCoA axes? Overall this analysis does not indicate that pH and salinity was responsible for the differences observed in community composition, but rather indicates that there are significant differences in community structure among sampling sites. Given the data in Table 1, these differences could also be attributed to TOC or N, so be careful not to assign attribution unless you have something specific confirming that. Without more info on how this analysis was run it is difficult to properly determine which feature of these sites are contributing to this change.
o. L239-244: How exactly is “top 50” defined here? The top 50 most abundant taxa from across all sites? This is not clear, so I am not sure how to assess the effectiveness of this analysis. What method was used to create these clusters (kmeans, hclust, etc.)? Do you see any issues with moving from Phylum-level to Genera-level analyses? Further, how did you assign functional classifications?
p. L255-256: Does “clearly varied” indicate statistical significance? Can this variation be quantified and expressed as a p-value, effect size, or some other quantifiable indicator of difference?
q. L261-267: Are any of the environmental variables correlated with one another? Could this potentially confound your interpretation of the impact of environmental variables?
r. L290-291: What exactly is “ecological environment factors” referencing that was not included in the previous two sentences?
s. L292:294 – Can you define ecological pressure in this context? There does appear to be a spatially distinct environmental gradient within the lake, but is this a natural feature or something that is the direct consequence of a human presence? If natural it is more of a barrier than a pressure as it delineates smaller communities within the lake. If it is unnatural, it may serve as a pressure if it is altering how those communities are distributed (see Carlson et al., 2019; Newton et al., 2011; Huang et al., 2019).
t. L318: “most primary” is redundant
u. L345-346: Is this referring to an environmental quality index or something similar? If so, can you provide a citation?
v. L452-454: Is there a citation you can point readers to for this claim? It was not previously discussed and should perhaps be included in your results.


Carlson et al., 2019 - https://www.nature.com/articles/s41396-018-0328-1
Custodio et al., 2022 - https://www.mdpi.com/2073-4441/13/16/2256
Huang et al., 2019 - https://www.ncbi.nlm.nih.gov/pmc/articles/PMC6391271/
Ma et al., 2022 - https://www.mdpi.com/1660-4601/19/21/13888
Newton et al., 2011 - https://www.ncbi.nlm.nih.gov/pmc/articles/PMC3063352/
Philippot et al., 2021 - https://journals.asm.org/doi/full/10.1128/MMBR.00026-20
Shade et al., 2012 - https://www.frontiersin.org/articles/10.3389/fmicb.2012.00417/full
Walker et al., 2022 - https://www.nature.com/articles/s43705-022-00162-z
Wang et al., 2016 - https://link.springer.com/article/10.1007/s12583-015-0618-8

---

## Round 0.2 · accepted · Accept

Congratulations.,. Your submission has been accepted for publication in PeerJ

Reviewer 1 ·

Basic reporting

This work aims at describing the bacterial community within an alkaline and saline lake in Ordos’ desert plateau of inner Mongolia. In order to do so, they sampled 4 points along the lake, following an environmental gradient of increasing salinity/pH. They have provided raw data, deposited in the SRA website. Figures and tables have good quality and are well described in the text, coinciding with the results.

My previous concerns about the quality of the manuscript ranged form poor conection/flow among samples, use of references/discussion outside of the scope of the results provided and shallow description about how changes in the microbial composition along the lake matches environmental factors.

I am happy to see that all comments were addressed, and the overall quality of the paper improved. I, therefore, recommend this paper for publishing.

Experimental design

Nothing to add. Discussed in basic reporting.

Validity of the findings

Nothing to add. Discussed in basic reporting.

Additional comments

Nothing to add. Discussed in basic reporting.